# A Study on the Improvement of Filter Performance to Remove Indoor Air Pollution

**Yong-Sun Kim [1], Hong-Gun Kim [1], Lee-Ku Kwac [2] and Sang-Cheol Ko [2,\*]**

1   Institute of Carbon Technology, Jeonju University, 303 Cheonjam-ro, Wansan-gu, Jeonju-si 55069, Jeollabukdo, Republic of Korea
2   Graduate School of Carbon Convergence Engineering, Jeonju University, 303 Cheonjam-ro, Wansan-gu, Jeonju-si 55069, Jeollabukdo, Republic of Korea
\*   Correspondence: scko@jj.ac.kr; Tel.: +82-63-220-2623

**Abstract:** This study carried out a simplified baffle filter shape study on the over the range (OTR) filter used in a general kitchen. In order to improve the filter's efficiency, the simulation was performed using ANSYS FLUENT and COMSOL Multiphysics, and a wind tunnel test apparatus was manufactured to minimize the error rate of dust collection efficiency in the experiment. In the case of a physical filter, it was confirmed through a theoretical review that dust is collected in the filter by the inertial collision method, and the flow velocity must be increased to increase the dust collection efficiency. To increase the flow velocity and increase the filter contact area, the use of sub-filters and the Coanda effect was proposed and simulated. When only the Coanda effect was applied, the collection efficiency increased by about 7–15% compared to the original filter, and when the three types of sub-filters were proposed, and among them, a circular sub-filter was applied, it increased by 25%. When applying the sub-filter and the Coanda effect at the same time, it was confirmed that the sub-filter was more efficient than the Coanda effect. However, in the case of a physical filter, since it cannot collect particles less than PM2.5, the electric dust collection method was proposed and a simulation was conducted. The possibility of removing ultrafine dust below PM2.5 was secured by using an electric dust-collection filter simulation, and it is expected that the reliability will be secured by using experimental devices and products in the future.

**Keywords:** indoor air pollution; filter; Coanda effect; collection efficiency

## 1. Introduction

Indoor air quality has a very important influence when it becomes difficult to go outdoors. There are many sources of indoor air pollution. When food is cooked, a large number of pollutants are generated in a short time. In general residential spaces, most of the structures are connected to the kitchen and the living space, so the hood plays an important role in preventing the diffusion of indoor pollution and properly discharging it to the outdoors. The filter mounted on the hood is an important research topic [1–5].

The hood filter has a great influence on the pressure loss and collection efficiency, depending on the shape. The typical shapes used for the filter are the mesh-type and the baffle-type [6–8]. A mesh-type filter is suitable for home use because it has low pressure loss and noise and uses a small-capacity fan. The baffle-type filter has the advantage of high collection efficiency; therefore, it is advantageous to use in restaurants and factories. However, the pressure loss is high, making the vibration large and the fan capacity increased [6–12].

The study of filters has been conducted in various ways using computational fluid dynamics (CFD) [3,4,13,14]. One type of baffle-types, the chevron-type filter, has high efficiency, but the pressure loss is also very high. Table 1 shows the shapes of generally used hood filters. The mesh-type filter is widely used for home use. It is less efficient but less noisy, so it is suitable for indoor use. Baffle-type filters and complex-type filters

are generally used for restaurants and industrial purposes. This is because restaurants or industrial sites that generate a lot of oil-mist and dust require high efficiency even if some noise is generated. The cyclonic-type filter has the advantage of high collection efficiency. However, the disadvantages are that it generates a lot of noise and vibration due to high pressure loss, and it is difficult to design for everyday use [14,15].

**Table 1.** Hood filter type and characteristics [14,15].

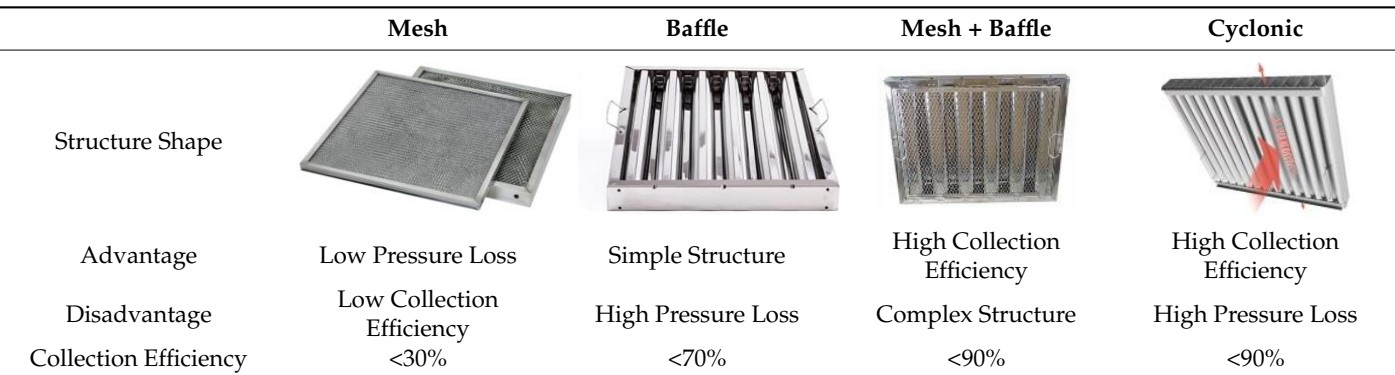

|  | **Mesh** | **Baffle** | **Mesh + Baffle** | **Cyclonic** |
|---|---|---|---|---|
| Structure Shape | | | | |
| Advantage | Low Pressure Loss | Simple Structure | High Collection Efficiency | High Collection Efficiency |
| Disadvantage | Low Collection Efficiency | High Pressure Loss | Complex Structure | High Pressure Loss |
| Collection Efficiency | <30% | <70% | <90% | <90% |

Kim, H. W. reported in their master's thesis that the chevron filter had a collection efficiency of 93% and a pressure loss of 6.35 mmAq [14]. Based on the previously studied shape, the structure is simplified to reduce the pressure loss and study the shape with high dust-collection efficiency.

According to Kim, H. W., the simulation technology was secured by copying the shape of the thesis [14]. A preliminary simulation was conducted by removing the protrusion to reduce the high-pressure loss, and a simple filter was proposed by removing the shape of the high-pressure section and selected as the basic shape.

In order to increase the contact time of the physical filter, the Coanda effect was applied. The Coanda effect is a phenomenon in which a fluid flows along a wall due to the surrounding pressure and the viscosity of the fluid when it flows along a curved surface. When the fluid flows along the wall and deviates from a certain pressure, flow separation occurs. By using the Coanda effect, the contact time of the filter was increased to improve the collection efficiency. Using this phenomenon, various home appliances have been released [16–20].

However, the Coanda effect alone cannot dramatically improve the collection efficiency. Therefore, by applying the shape of the sub-filter under the main filter, the efficiency according to the shape is compared, and an optimal filter improvement is proposed. The purpose of this study is to propose an optimal filter with high efficiency and low pressure loss, and to secure the reliability of the simulation through the experiment.

To select the optimum filter shape and secure reliability, wind tunnel device fabrication and experiments are conducted to compare the simulation results with the experimental results to achieve an error rate of 10% or less.

In addition, as a result of simulation and experimentation, a physical filter cannot remove ultrafine dust less than PM2.5, so we simulate an electrostatic precipitating filter to remove the dust of a size that cannot be removed from the physical filter.

## 2. Simulation Method and Result

Simulations were conducted on physical filters, i.e., sub-filters and Coanda effects, using ANSYS FLUENT 19.0. The multiphase flow of particles and air was simulated and $k - \varepsilon$ was used as the turbulent component. In addition, a simulation was conducted on the electric dust collection filter using COMSOL Multiphysics v5.5. Multiple physical analyses of particle-flow–electricity were conducted, and $k - \varepsilon$ was used as the turbulent component as well as the physical filter. The Coanda effect simulation is performed first, and then the simulation is performed by applying a sub-filter. Two simulations were conducted to

simulate the optimal shape in a complex way, and in order to secure the reliability of the simulation, an experiment was conducted and compared after the shape was fabricated. In the case of the wind tunnel device, it was manufactured using acrylic, and the number of particles before and after the filter was measured using a particle counter to evaluate the dust-collection efficiency. The flow of this simulation is shown in Figure 1.

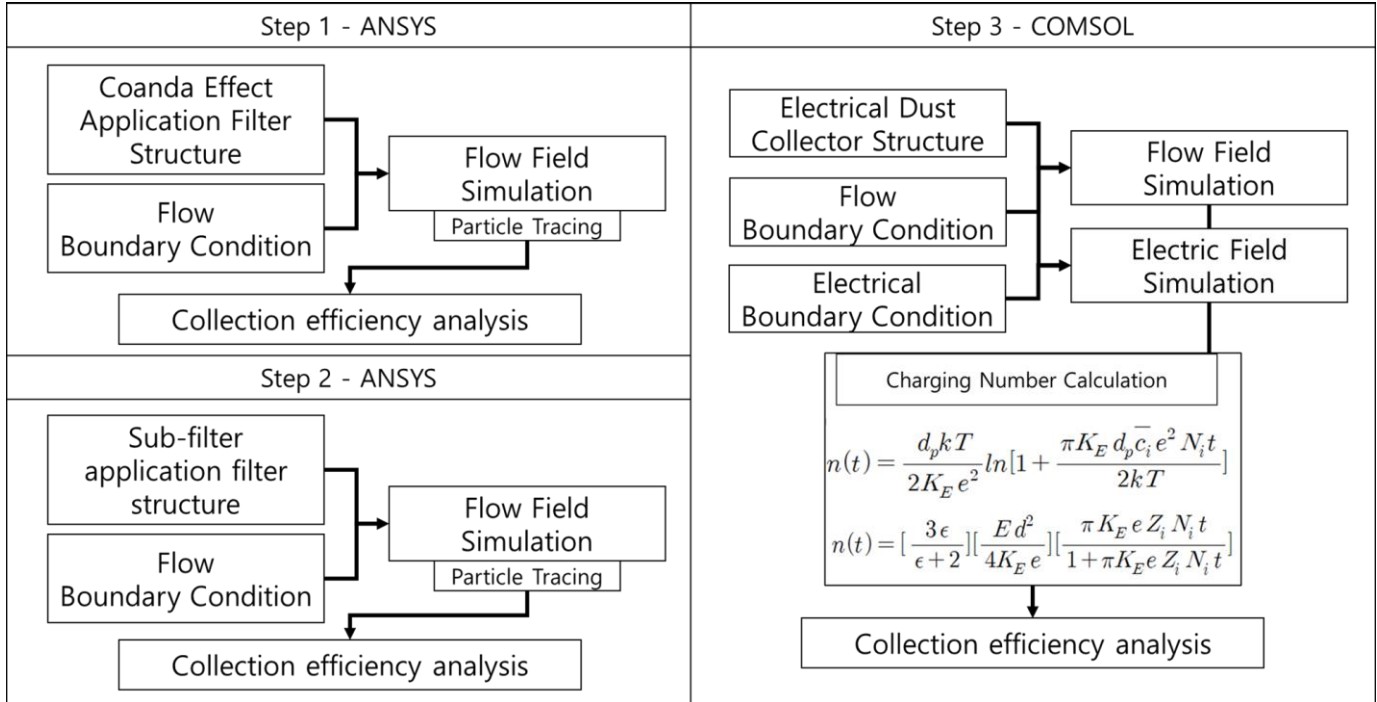

**Figure 1.** Physical and electrical filter simulation flow.

### 2.1. Coanda Effect Simulation

Figure 2 shows the boundary conditions of the simulation to apply the Coanda effect. The thickness of the filter is 2 mm, and the gap between the filters is 2 mm. The total height h of the main filter was modeled by setting it to 30 mm. The fluid flows upward from under the filter, and the flow rate of the inlet boundary condition was set to 1 m/s.

In the case of the slide model in Figure 3b, a tap was applied by moving 0.5 mm in the +x direction to form a low-pressure section in the filter shape. In the case of the bottom model in Figure 3a, an oval shape of long axis Φ 3 and short axis Φ 2 was added to the bottom of the filter to form a low-pressure section. The shape of the slide model and the bottom model is shown in Figure 3.

### 2.2. Coanda Effect Simulation Result

In the simulation, a multiphase flow was performed to express both particles and air at the same time. The total inflowing particles are collected by the filter by spraying 10 k particles from the inlet, and collected by the filter, and the uncollected particles are discharged to the outlet. The filter's dust-collection efficiency equation is shown in Equation (1).

$$\text{Efficiency}(\%) = \frac{\text{Trap particle}}{\text{Total paticle}} \tag{1}$$

Table 2 shows the filter dust collection efficiency, pressure loss, and maximum speed values according to the shape. In the case of the filter to which the Coanda effect is applied, the efficiency improvement of 7~15% compared to the original model can be confirmed. As a result of the analysis, the dust collection efficiency was high in the order of original < bottom < slide model. When the pressure loss of the filter is less than 3.5 mmAq,

there is relatively little noise and vibration, so it can be used indoors. The proposed filter satisfies the maximum pressure loss.

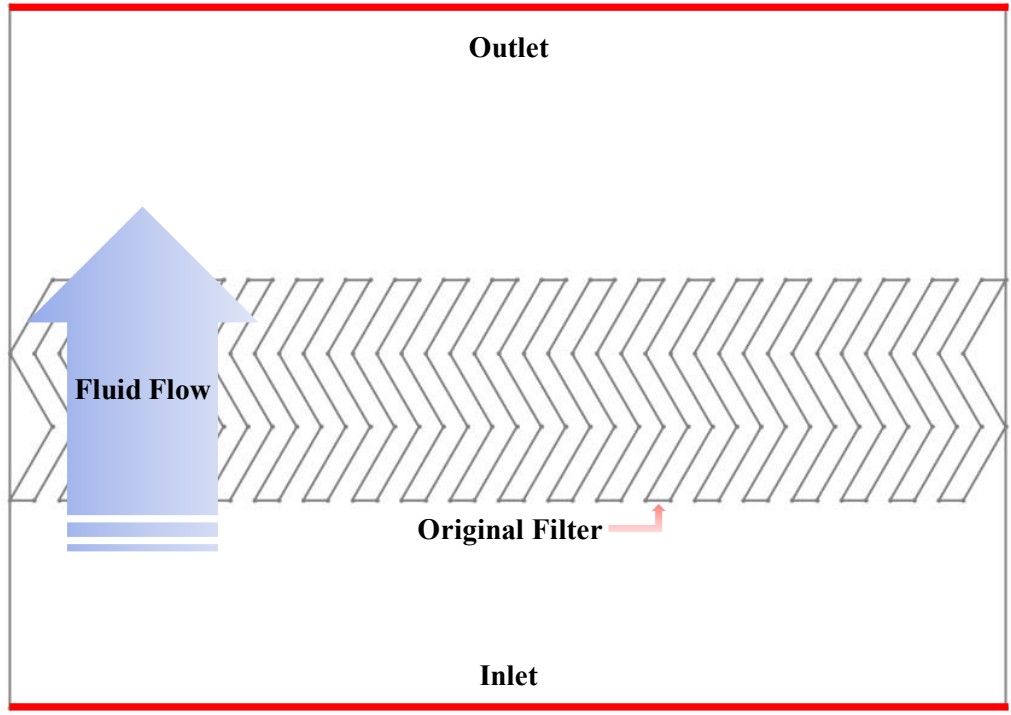

**Figure 2.** Boundary condition of original model.

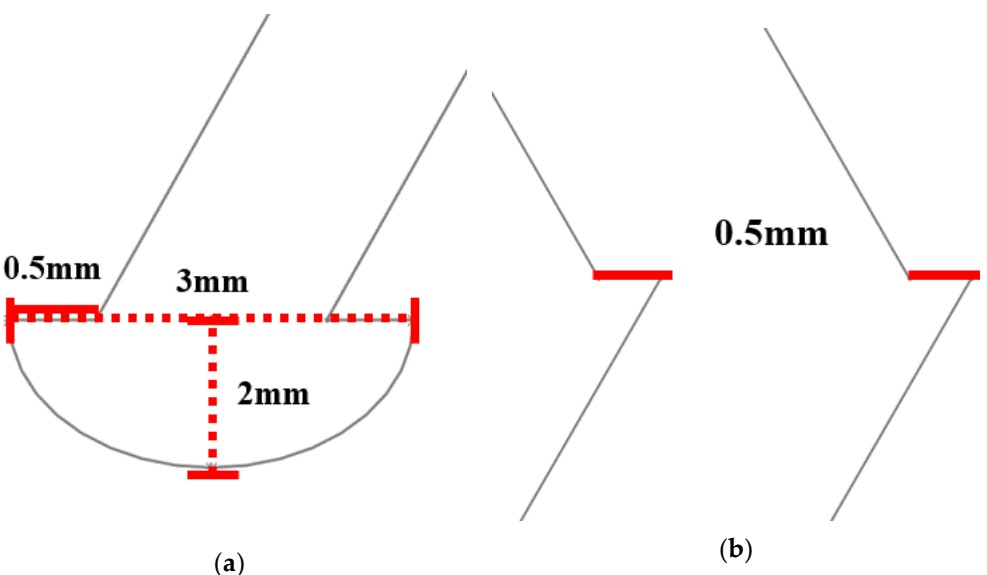

**Figure 3.** Shape of (**a**) bottom model and (**b**) slide model.

**Table 2.** Coanda filter-type dust-collection efficiency, pressure loss, and maximum flow rate.

| Type | Trapped Particle (ea.) | Efficiency (%) | Pressure Loss (mmAq) | Velocity (m/s) |
|---|---|---|---|---|
| Original | 1541 | 15.41 | 1.29 | 4.35 |
| Bottom | 2297 | 22.97 | 1.72 | 5.23 |
| Slide | 3051 | 30.51 | 1.72 | 5.38 |

When the particles move along the streamline in the filter, they are collected by the filter through the inertia of the particles. This action is called inertial collision, and it can be expressed, as in Equation (2), by the Stokes number as the ratio of the particle stopping distance to the filter diameter [17–21].

$$S_{tk} = \frac{\tau U_0}{d} \tag{2}$$

Here, $\tau$ is the relaxation time until the particle leaves the streamline and returns to the streamline again, which is calculated by Equation (3) [17–21].

$$\tau = \frac{1}{18} \frac{d^2 \rho_P}{\mu} \tag{3}$$

$U_0$ is the initial velocity of the particle, and $\tau U_0$ is the shortest distance the particle has moved during the relaxation time. The larger $S_{tk}$ means that there are many colliding particles, and the larger this is, the higher the dust-collection efficiency of the filter.

Figure 4 shows the pressure distributions and velocity vectors of the three proposed filters. It is possible to visually confirm that low pressure is formed, and the back flow occurs near the tap to apply the Coanda effect. Figure 5 shows the dust-collection efficiency according to the particle size.

In the case of the Slide model, the dust-collection efficiency was increased by about 15% compared to the original model, and the Coanda effect helped to improve the efficiency. However, a filter with an efficiency of 30% has a low dust-collection efficiency when used as an indoor kitchen hood, and cannot collect particles of 3–7 μm in size, so it is ineffective in practice.

Therefore, to increase the collection efficiency of the filter, as suggested from the results of the simulation and experiment, a sub-filter was proposed as a turbulence generator to increase the flow velocity and contact time of the filter, and the simulation was conducted.

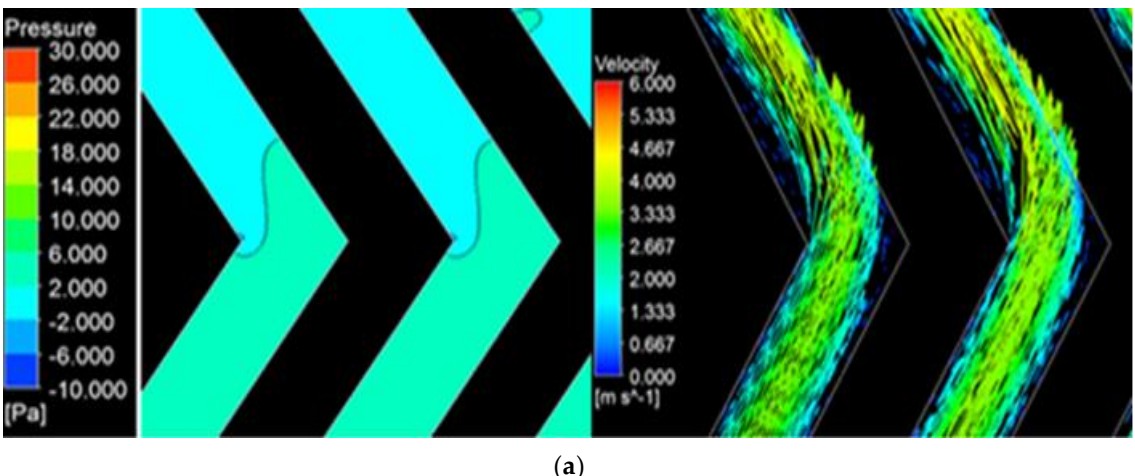

(**a**)

**Figure 4.** *Cont.*

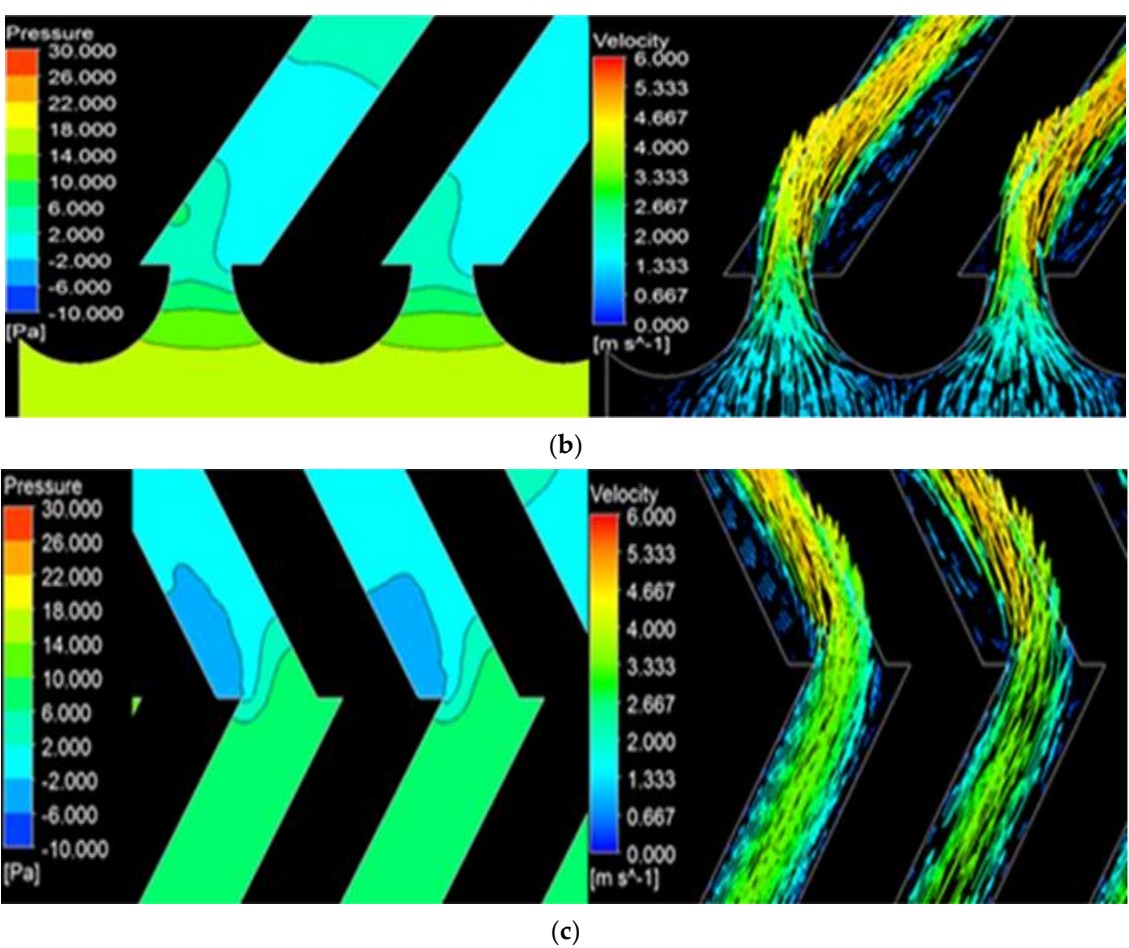

**Figure 4.** (**a**) Original; (**b**) Bottom; and (**c**) Slide model pressure and vector contours.

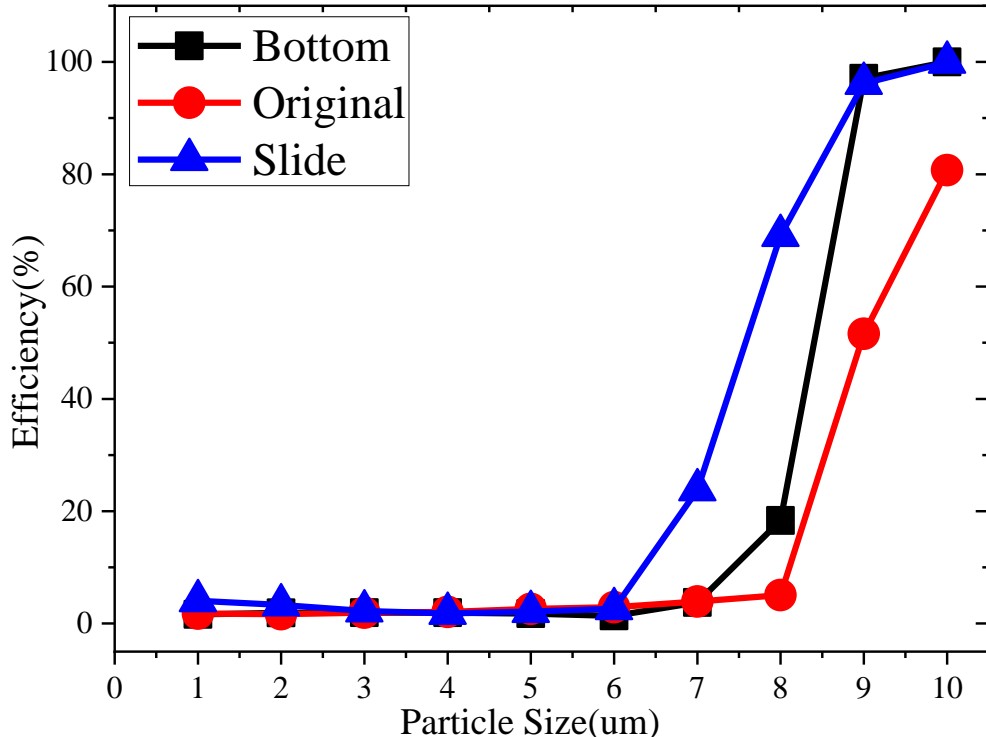

**Figure 5.** Efficiency by particle size in Coanda-applied filter.

### 2.3. Sub-Filter Simulation

Figure 6 shows the boundary conditions applied to the simulation to analyze the efficiency and pressure loss of the filter to which the sub-filter is applied. The fluid flows from the lower part of the filter to the upper part at a rate of 1 m/s. In addition, the circular-type, droplet-type, and cone-type sub-filters were selected, and the three shapes are shown in Figure 7. The sub-filter was installed in anticipation of the role of a turbulence generator that induces turbulence in the flow of the main filter.

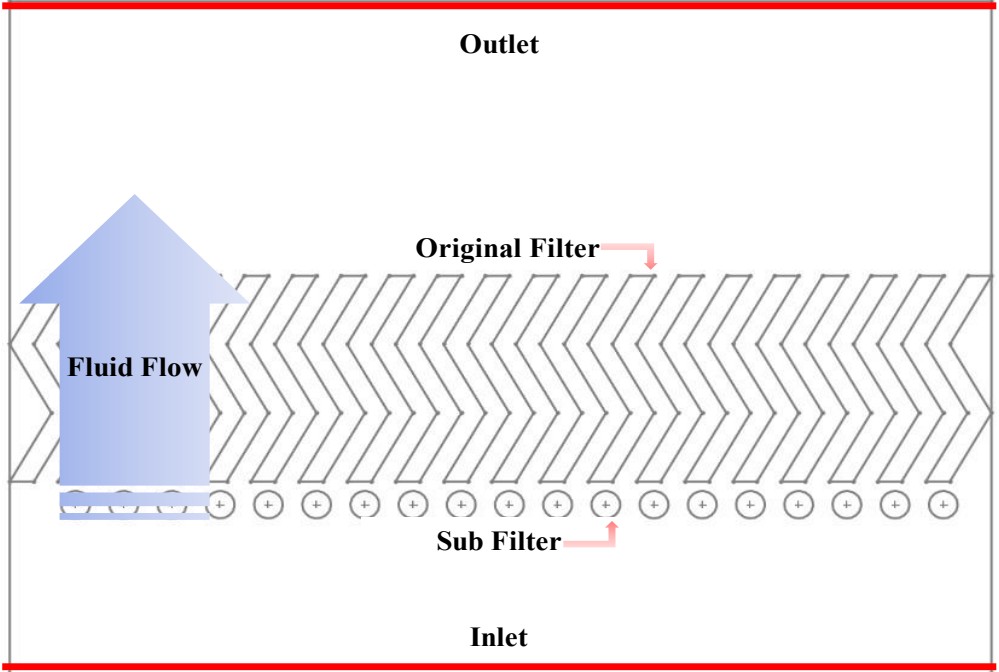

**Figure 6.** Analytical boundary conditions for sub-filters mounted under the filter.

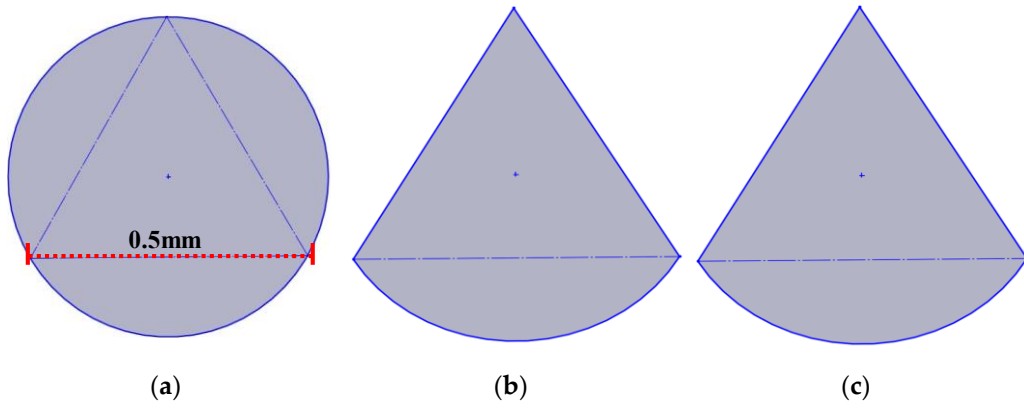

**Figure 7.** Three features proposed as sub-filters. (**a**) Circle; (**b**) Droplet; (**c**) Cone.

### 2.4. Sub-Filter Simulation Result

Figure 8 shows the pressure distribution and vector component of the filter. By examining the direction of the vortex generated in the wake of the sub-filter, it can be seen that the sub-filter acts as a turbulence generator at the entrance of the main filter. The increase in efficiency as the speed increases confirmed that the Stokes number increases in proportion to the speed in the physical filter dust collection of the inertial collision method.

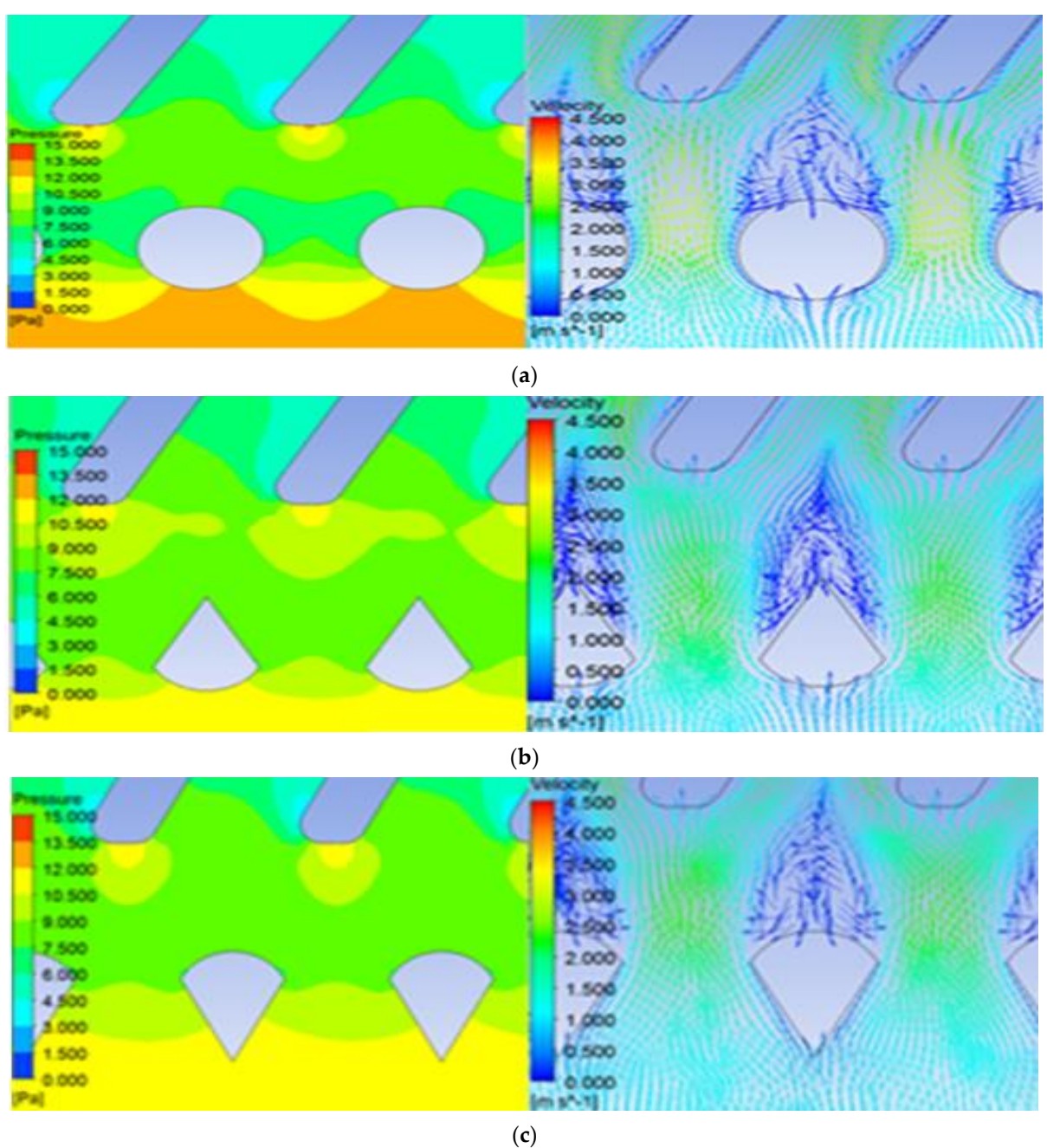

**Figure 8.** (**a**) Circle; (**b**) Droplet; and (**c**) Cone pressure and vector contours.

After applying the sub-filter to Table 3, the filter's dust-collection efficiency, pressure loss, and flow velocity inside the filter are shown Figure 8.

**Table 3.** Efficiency by sub-filter.

| Type | Trapped Particle (ea.) | Efficiency (%) | Pressure Loss (mmAq) | Velocity (m/s) |
|------|------------------------|----------------|----------------------|----------------|
| N/A | 3680 | 36.80 | 0.938 | 1.24 |
| Circle | 6409 | 64.09 | 1.261 | 1.45 |
| Droplet | 6378 | 63.78 | 1.142 | 1.44 |
| Cone | 6132 | 61.32 | 1.140 | 1.40 |

As a result of the analysis, by applying the sub-filter, the dust-collection efficiency was high in the order of circle > droplet > cone, and the pressure loss was up to 1.261 mmAq. Even when the sub-filter was applied, it did not have a significant effect on the pressure loss. It is recommended that the pressure loss required by general companies is less than 3.5 mmAq. This is to control noise and vibration. In this respect, the Coanda effect and the sub-filter satisfy the pressure-loss condition. The reason for the high efficiency of the circular shape is that the turbulent component of the circular sub-filter is higher than that of other shapes.

Figure 9 shows the efficiencies of the three proposed filters by particle size. The graph shows that the circular sub-filter has a good effect between 4 μm and 7 μm.

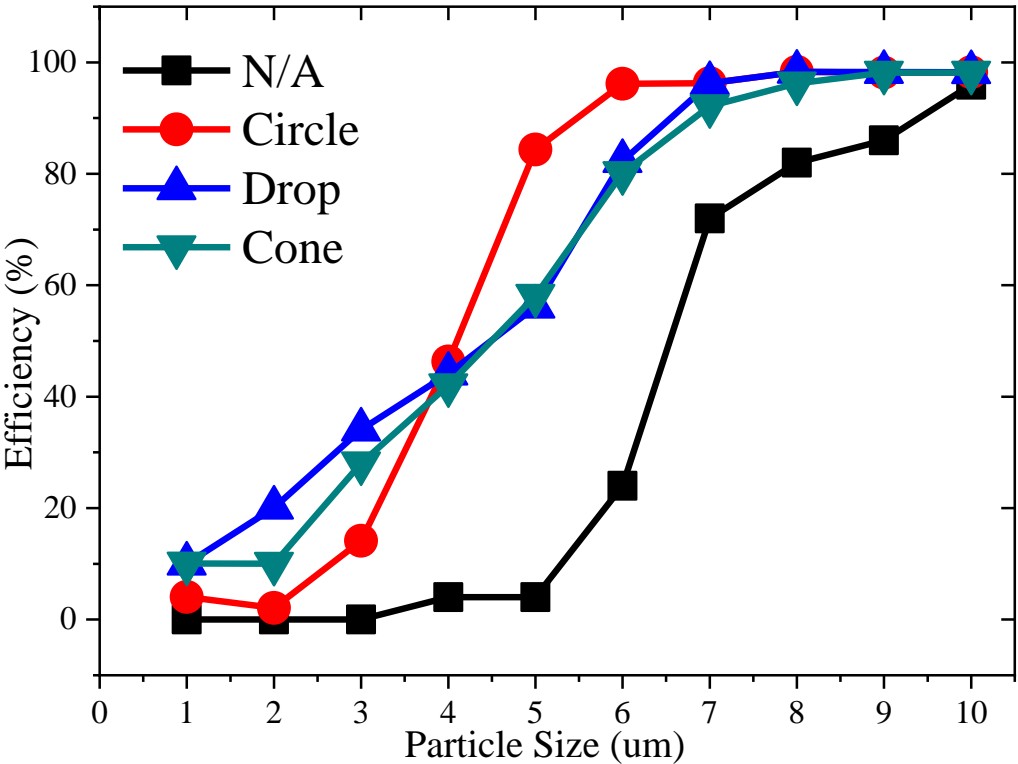

**Figure 9.** Efficiency by particle size with sub-filter applied.

*2.5. Apply the Coanda Effect to the Sub-Filter*

As a result of simulating the dust-collection efficiency in Section 2.4, in order to apply the Coanda effect to the circle filter with high dust-collection efficiency, the main filter was a 0.5mm slide-type filter, which we used to compare the dust-collection efficiency. Applying the sub-filter and the Coanda effect to the original filter increased the dust-collection efficiency from 15.41% to 68.6%. The dust-collection efficiency increased from 30.51% to 69.3% by applying the sub-filter of the slide filter to which the Coanda effect was applied.

Figure 10 is a graph comparing before/after attaching the sub-filter to the Coanda effect filter. When the Coanda effect is used alone, it can be confirmed that the effect is large, but if a sub-filter is installed, the effect of the Coanda effect on the dust-collection efficiency is small.

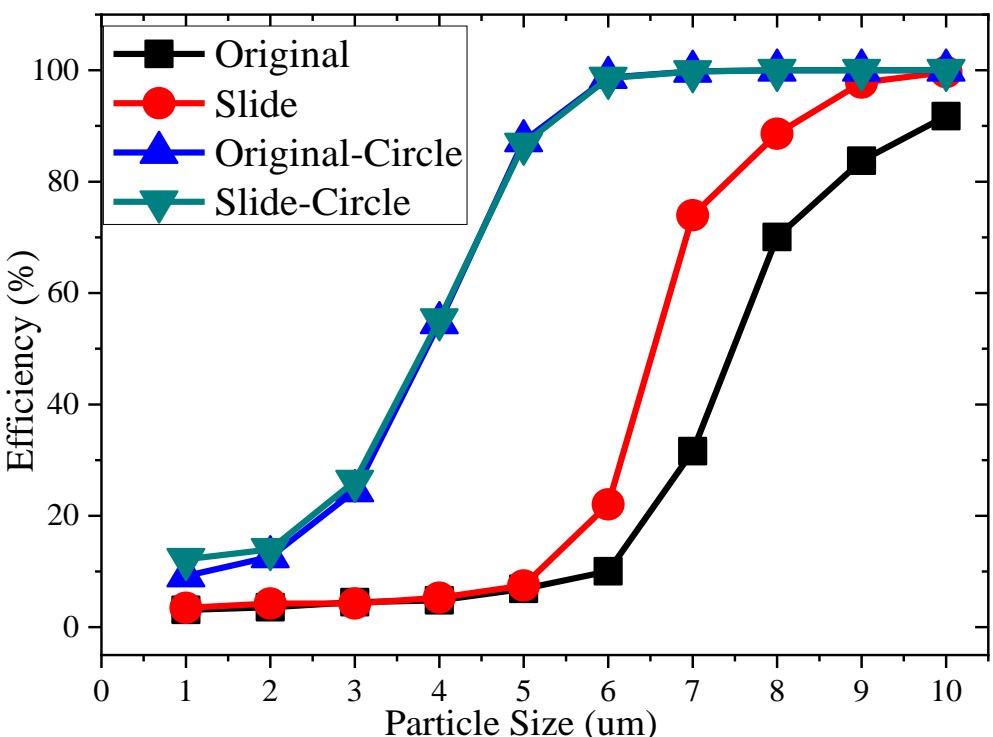

**Figure 10.** A comparison graph of filters with and without sub-filters.

*2.6. Experimental Equipment Design and Experimental Results*

For the simulation and comparison, the efficiency test equipment of the filter was designed as shown in Figure 11 and was manufactured using acrylic. The measurement method is as shown in Figure 12b,c. A filter manufactured using a 3D printer was installed, and the oil mist generated by the atomizer was inhaled using a Sirocco fan. The dust-collection efficiency was calculated by measuring the amount generated and the amount exiting the filter by using the particle counter installed in front and rear of the filter.

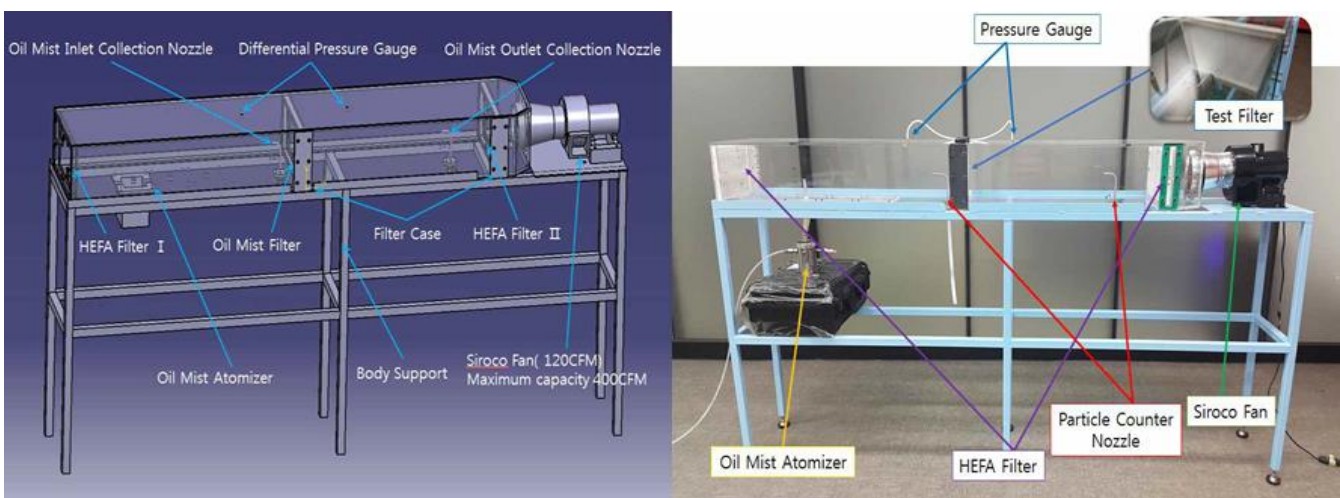

**Figure 11.** Design drawing and filter experiment device.

The sample particles used were Di-Octyl-Phthalate, and TROTEC CO. Fluke's PC-220 product was used as the particle counter. The particle counter channel can only measure 1 μm, 2.5 μm, 5 μm, and 10 μm sizes, where M stands for measurement and S stands for simulation.

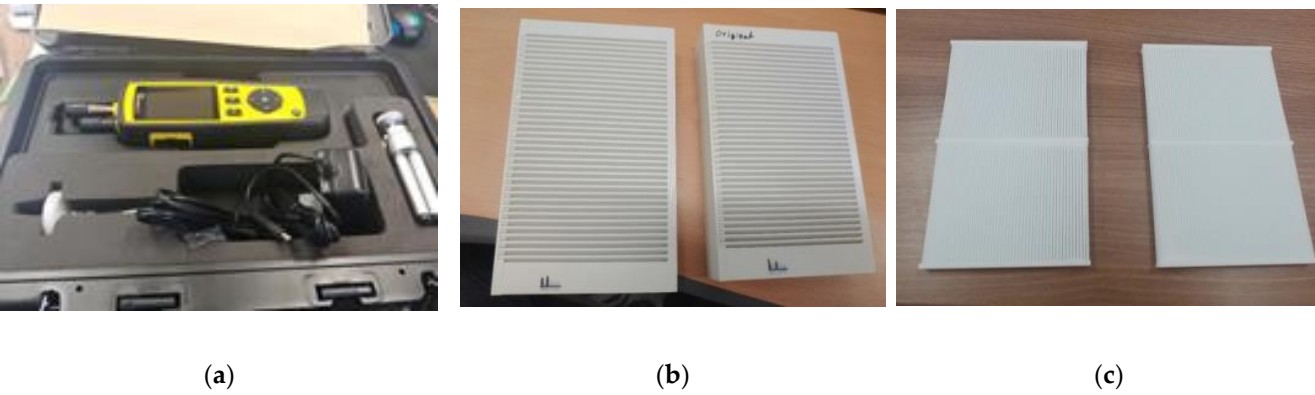

(**a**)                                        (**b**)                                        (**c**)

**Figure 12.** (**a**) Particle counter, (**b**) Coanda filter, (**c**) sub-filter.

Among the filters to which the Coanda effect was applied, an original model and a slide model were produced using a 3D printer and an experiment was conducted. Figure 13 shows the graph compared to the simulation. The error rate of the particle diameter suitable for the particle counter channel was measured and averaged. In the case of the original model, the error rate was calculated as 10.25%, and in the case of the slide model, the error rate was calculated as less than 10.07%.

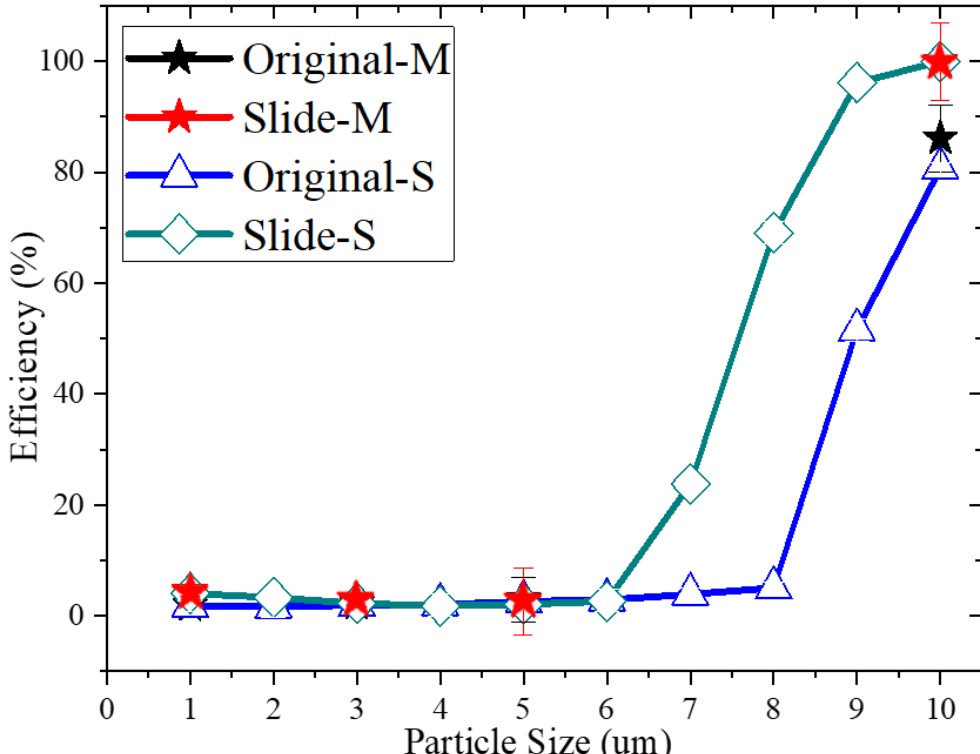

**Figure 13.** Comparison graph of filter with Coanda effect applied.

Among the filters to which the sub-filter was applied, the simplified filter and circle filter were manufactured and tested. Figure 14 shows the graph compared to the simulation. In the case of the circle filter, the error rate was 7.04%, and in the case of the S-filter, the error rate was 2.19%, ensuring the reliability of the simulation. In the future, if a light-scattering particle counter is used instead of an infrared-scattering-type particle counter, it is expected that the number of various channels can be measured. Then, since particles of various sizes can be measured, the reliability of the simulation will be accurately measured.

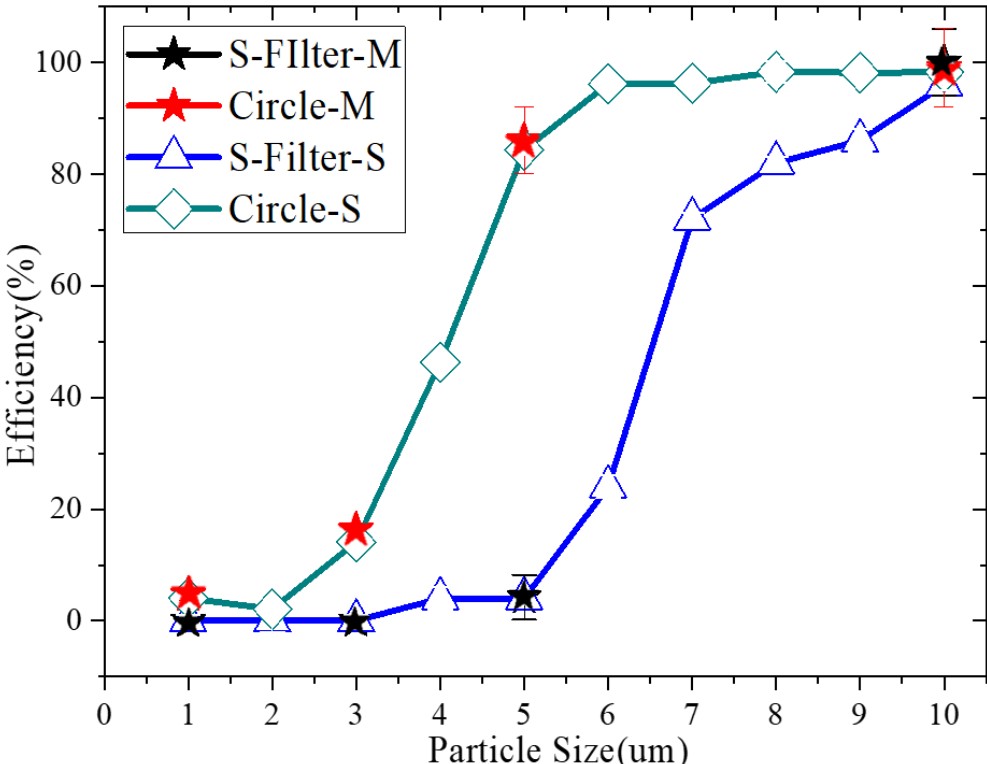

**Figure 14.** Comparison graph of filter with sub-filter applied.

*2.7. Electric Dust Collector Simulation to Remove the Dust under PM2.5*

In the case of the previously simulated physical filter, it can be seen that the dust-collection efficiency is lowered for particles of 5 μm or less. Therefore, it was simulated to collect particles of 0.1 μm to 5 μm or less using an electric dust-collection method. In order to calculate the dust-collection efficiency, a theoretical technology for the electric dust-collection method was secured.

Dust particles require a high ion concentration in the electric field to be sufficiently charged. When a voltage higher than the starting voltage is applied between the electrode and the ground, dust in the air is charged by a strong electric field. The principle of dust collection of an electric dust collector is that (+) dust charged by (+) electrodes is attached to the ground under the influence of repulsion by the (+) polarity of the filter [22–26].

Although there are various equations, the amount of electric charge applied to a single dust particle was selected by referring to Hinds' book. Hinds announced that there is both field charging and diffusion charging in electricity, and summarized it in an equation. This can be expressed by Equations (4) and (5). In the case of field charging, the larger the particles, the larger the charge, and in the case of diffusion charging, the smaller the particles, the larger the charge [22].

$$n_1(t) = \frac{d_p kT}{2K_E e^2} \ln\left[1 + \frac{\pi K_E d_p \overline{c_i} e^2 N_i t}{2kT}\right] \tag{4}$$

$$n_2(t) = \left[\frac{3\varepsilon}{\varepsilon + 2}\right]\left[\frac{Ed^2}{4K_E e}\right]\left[\frac{\pi K_E e Z_i N_i t}{1 + \pi K_E e Z_i N_i t}\right] \tag{5}$$

The variables used here are: $d_p$ is particle diameter, $k$ is the Boltzmann constant, $K_E$ is the electrical propositional constant, $\overline{c}$ is the average thermal speed of ions (240 m/s at standard room condition), $E$ is the intensity of the electric field, $\varepsilon$ is the relative permeability of KCl, $Z_i$ is the mobility of ions, and $N_i$ is the concentration of ions. Most are specified in units commonly used in the SI unit system among the variables; $E$ and $N_i$ can be inferred

from the results of the electric field simulation. The study simulated a limited space, and the flow rate is 1 m/s, so the variable *t* is fixed. Here, *t* means the time during which the dust particles are charged. The charging number applied to one dust particle was obtained by using Equations (4) and (5), with the result values obtained from these variables and the simulation [22,23].

If this electric charge is combined, a combined area can be seen, as illustrated in Figure 15, and it can be confirmed that a section of 0.3 μm is the most extreme environment. These data are extracted from references and Hinds books [22,27,28]. This is currently the standard for setting the test particle to 0.3 μm in the Korea Air Purifier Association's Indoor Air Purifier Standard (SPS-KACA002-132) [29].

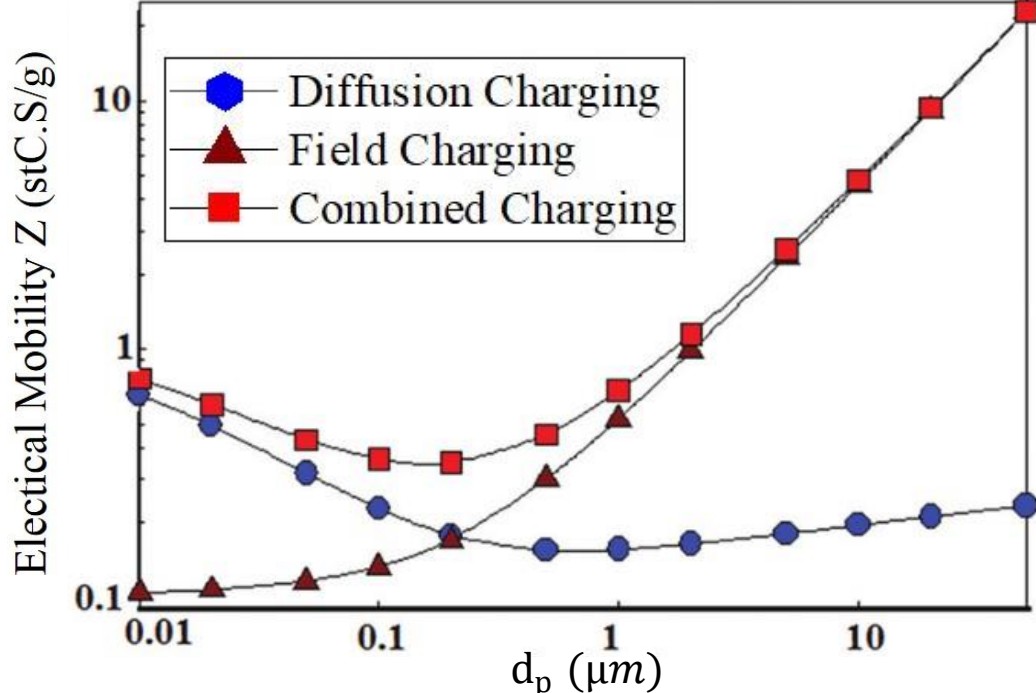

**Figure 15.** The combined charging is a function of the particle diameters.

Figure 16 shows the boundary conditions of the electric dust collector filter simulation. The simplified chevron filter mentioned in Figure 2 was used. Electrodes and grounding were added to the lower part of the filter to generate static electricity. Dust can be charged from the generated static electricity, and the flow escapes from the lower part of the filter to the upper part. A total of 8 kV was applied to the inlet electrode and 5 kV was applied to the dust filter to push the charged dust particles by repulsion.

In this study, COMSOL Multiphysics v5.5 was used for the multi-physical analysis for the electric dust collector simulation. Flow analysis, electric field analysis, and particle tracking were performed simultaneously. As with the existing ANSYS simulation, 10 k dust particles were dispersed at the entrance to determine the number of traps to the electric dust-collection filter, and the dust-collection efficiency was calculated. Figure 17 shows the particle tracking result of a simplified I-type filter, which shows that charged particles from the electrodes are collected in a dust filter.

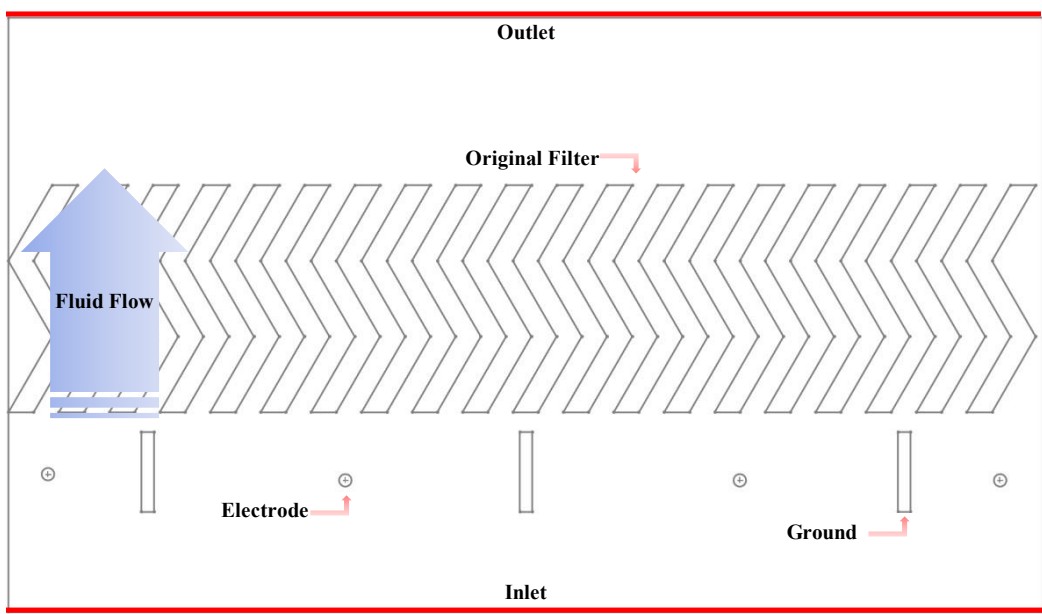

**Figure 16.** Boundary condition of electric dust-collection S-filter.

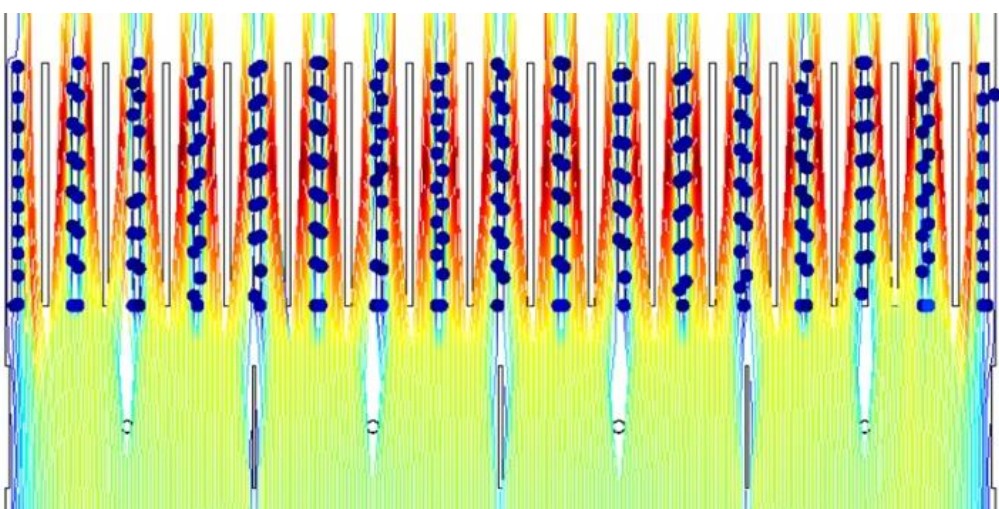

**Figure 17.** Particle tracking contour of I-filter.

Figure 18 shows the collection efficiency depending on particle sizes, and it can be seen that the efficiency increases as the particle size increases. At 0.1 μm, even if the diffusion-charge and field-charge values were high, the charging number was low because the particle size was small, and at a size of 2 μm or more, the charging number was high enough to trap the particles, so the efficiency was 98% or more. In addition, the difference in dust-collection efficiency between the existing S-filter and the simplified form of the I-filter is insignificant, and it is judged that the shape of the filter can be simplified in the electric dust-collection type filter.

The simulation used in this research was conducted in light of the charging number according to the particle size, and the applied charging number was calculated using the formula, so some assumptions were made. If calculated using the user definition function (UDF) to supplement these, it is judged that a more accurate result can be obtained.

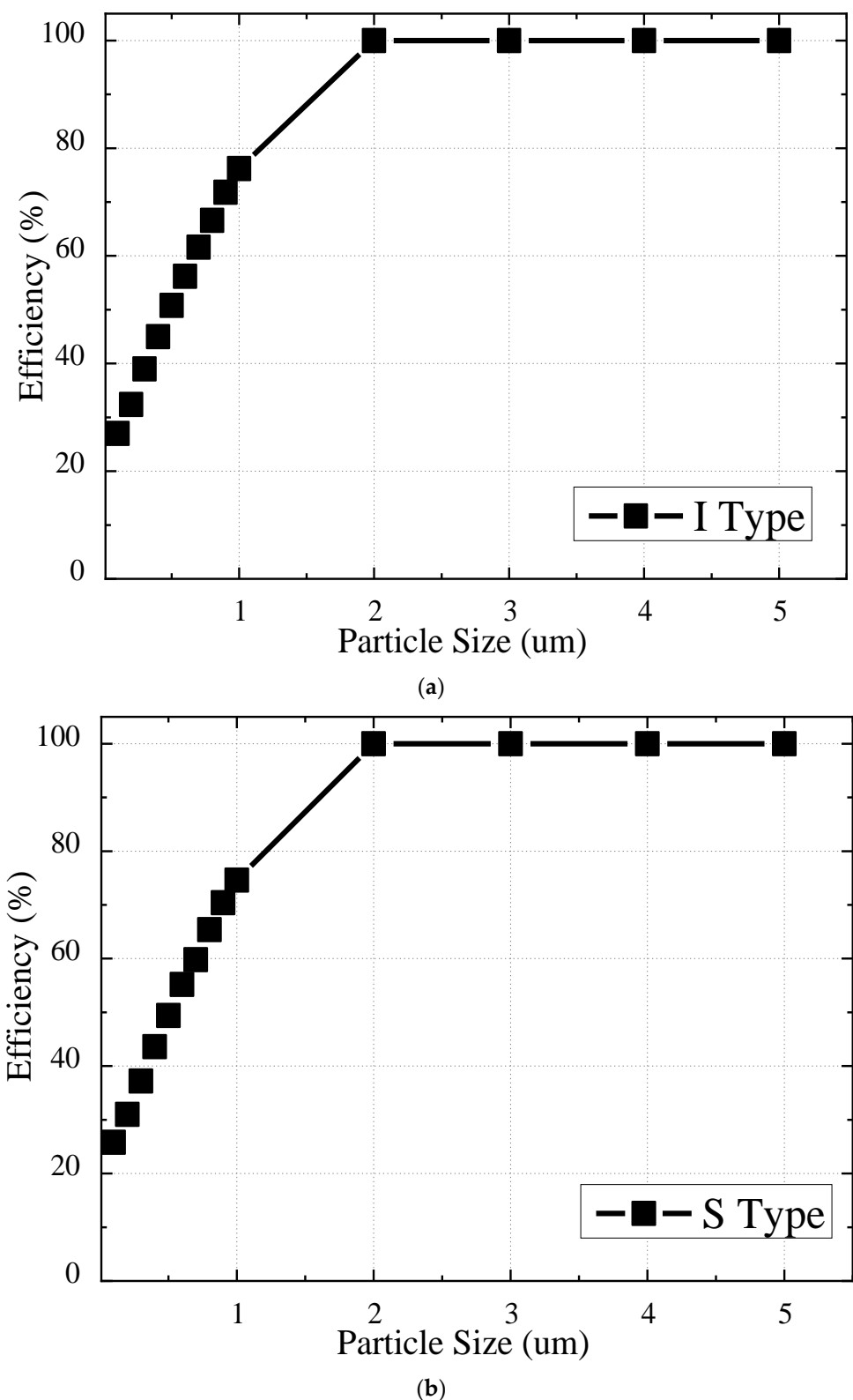

(**a**)

(**b**)

**Figure 18.** Efficiency comparison of I-Filter (**a**) and S-Filter (**b**).

## 3. Conclusions

A shape study was conducted on a physical over the range (OTR) filter used in general kitchens. A shape was proposed to improve the efficiency of the filter through simulation; physically proposed filters have the advantage of reducing pressure loss, and electrically

proposed filters have the advantage of collecting ultrafine dust. The conclusions drawn from the experimental results can be summarized as follows.

1.  A simple baffle-type filter was selected as the basic filter by removing protrusions that cause high pressure loss from the previously studied chevron-type filter. The Coanda effect was applied to the basic baffle-type filter to adjust the flow rate and allow dust particles to flow on the wall surface of the dust collection filter. As a result, when the Coanda effect was applied, the efficiency increased by 7 to 15%. However, it is very inefficient to use in a typical kitchen.

2.  In order to further increase the effect of the existing filter, a sub-filter was installed at the lower part of the main filter to serve as a turbulence generator. After simulating three types of filters, circular type, conical type, and droplet type, the circular-type sub-filter worked the best. The dust-collection efficiency is not much different, but it is increased by about 25%. These results show that sub-filters have a greater effect on efficiency than filters with the Coanda effect.

3.  The simulation should ensure reliability throughout experiments. A wind tunnel test device suitable for the standard test method of the Korea Air Purifier Association was designed and manufactured. A filter used in the test was manufactured using a 3D printer, and particles were sprayed using di-Octyl-Phthalate. As a result, the filter to which the Coanda effect was applied achieved an error rate of about 10% compared to the simulation, and the filter to which the sub-filter was applied showed an error rate of about 2–7%. This demonstrates that the results of the simulation are somewhat reliable.

4.  An electrostatic dust-collection filter was proposed to remove ultrafine dust that is difficult to remove with a physical filter. It was confirmed that ultrafine dust below PM2.5 can be removed. In addition, since there is no difference in the dust-collection efficiency according to the shape of the dust-collection filter, structural simplification is also possible. In the future, it is expected that a more realistic result will be obtained if the simulation is conducted so that it can be charged in real time according to the movement of the particles. In addition, we intend to secure the reliability of the simulations by manufacturing an actual electrostatic dust collection filter. The experimental device is also equipped with a sealed chamber for testing the purifying capacity system based on the test standards of the Korea Air Purifier Association, and thus it is judged that the reliability of the experiment can also be increased.

**Author Contributions:** Conceptualization, H.-G.K., S.-C.K. and L.-K.K.; methodology, L.-K.K., S.-C.K. and H.-G.K.; validation, Y.-S.K. and S.-C.K.; software, Y.-S.K.; formal analysis, S.-C.K.; investigation, Y.-S.K.; resources, Y.-S.K. and S.-C.K.; data curation, Y.-S.K.; writing—original draft preparation, Y.-S.K.; writing—review and editing, Y.-S.K. and S.-C.K.; visualization, Y.-S.K.; supervision, S.-C.K.; project administration, H.-G.K.; funding acquisition L.-K.K. and H.-G.K. All authors have read and agreed to the published version of the manuscript.

**Funding:** This research was also supported by the Basic Science Research Program through the National Research Foundation of Korea (NRF) funded by the Ministry of Education (No. 2016R1A6A1A03012069). This work was supported by the National Research Foundation of Korea (NRF) funded by the Ministry of Education (NRF-2017R1D1A1B03036070).

**Institutional Review Board Statement:** Not applicable.

**Informed Consent Statement:** Not applicable.

**Data Availability Statement:** Not applicable.

**Conflicts of Interest:** The authors declare no conflict of interest.

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
