# Peer review of "A Study on the Improvement of Filter Performance to Remove Indoor Air Pollution"

_applsci, doi:10.3390/app13042561_

Round 1

Reviewer 1 Report

In this manuscript, Yong-Sun Kim, et al perform a systematic simulation study on the filter efficiency of the over the range (OTR) filter. It was shown that applying Coanda effect into the filters effectively increases the dust collection efficiency by the inertial collision of the particles. By adding sub-filter to the main filter, which acts as a turbulence generator at the entrance of the main filter, the efficiency is further increased. The efficiencies obtained from the simulation are directly compared with the experimental results, which shows good simulation reliability. It is further demonstrated that by introducing electrostatic dust collector, the collection efficiency on small dust with diameter smaller than 5 um is effectively increased. Overall, it is a well-done work with detailed simulation study and experiments, which provides an important insight about the directions for developing novel filters with improving dust collection efficiency. Therefore, I suggest minor revision with the following comments:

1)     In Figure 8, the efficiencies of different types of sub-filter are compared. The author should discuss why the circle shape exhibits a stronger practice size dependent efficiency compared to other types.

2)     The author should increase the quality of the figures. In some figures, like fig.14, it is really difficult to read the description in it.

3)     There are some typos and grammar mistakes in the current manuscript, the authors should carefully proofread and correct it. For example, “In 3.4, in order…” on line 185, page 7. “Fig. Fig.14…” on line 241, page 10. “Structure When applying…” on line 269, page 11.

Reviewer 2 Report

 Article

A Study on the Improvement of Filter Performance to Remove Indoor Air Pollution

General Comments:

1)    The manuscript is nearly a primary report, not a scientific research!!

2)    The manuscript in need for editing completely.

3)    The manuscript in need for extensive English editing.

4)    The flow and harmony of the manuscript need to be enhanced and modified completely.

5)    What do you mean by a shape study (line 10)?

6)    Please, specify the error rate (for what parameters) (line 13)?

7)    The abstract need more clarification.

8)    The first paragraph of “Introduction” (lines 28-30), must be reference and edited in a scientific form?

9)    Pollution indoors changed to indoor pollutants (line 33).

10) (lines 36-37): “The typical shape used for the filter is the mesh type and the 36 baffle type”: must be must be reference and edited in a scientific form?

11) Abbreviation “CFD” in line (41) must be identified.

12) Line (43): “It was reported that the Chevron filter in the previous research paper” what do you mean by “previous research paper”?

13) Table (1): Please insert a new column for the references of these information?

14)     (lines 50-51): what do you mean by “previous research paper”?

15)     (lines 53-54): the coanda effect must be described well!!

16)     The authors describe the modification process by two parameters (reduction of pressure loss, and increase the contact time of the physical filter? why the two parameters only were selected for the modification process of the indoor filter efficiency?

17)     The main objective of the manuscript is not specific and deterministic?

18)     (lines 67-71): the paragraph must be transported to the “Methodology” section.

19)     For the section “Simulation Method and Results”, the authors must insert a paragraph describing the next subheadings.

20)     (line 73): The symbol “&” must be deleted.

21)     The manuscript need a clarification chart for the used methodology.

22)     Line (78): “In the case of the slide model”, what do you mean by this model?

23)     Line (79): “In the case of the bottom model”, what do you mean by this model?

24)     The authors must add (trapped particles=total particles-discharged particles).

25)     Line (100): “All the proposed filters have satisfactory results”, what do you mean?

26)     Title of “Table 2” must be detailed?

27)     Line (99), what about the measured levels of “noise and vibration” for the three types of filters?

28)     What about the measured levels of “trapped dust” for the three types of filters?

29)     What about the measured levels of “discharged dust” for the three types of filters?

30)     What about the measured levels of “total dust” for the three types of filters?

31)     What is the applied instrument for measuring the dust?

32)     Is the applied instrument for measuring the dust is calibrated!

33)     Lines (128-131): this paragraph must be transferred before “Figure 3”.

34)     In Figure (3), where the direction of flow?

35)     Lines (140-141): the paragraph must be detailed and clarified?

Round 2

Reviewer 2 Report

1. The required extensive English editing not considered the benefits, as replied.

2. Lines (46-53) must be referenced

3. The manuscript need a flowchart to show the applied the methodology

4. Lines (66-69) must be referenced

5. (ANSYS FLUENT 19.0) must be detailed and referenced

6. (COMSOL Multiphysics v5.5) must be detailed and referenced

7. The thickness of the filter is 2t (line 96), what is "t"?

8. lines (120-121) Controlling the noise and vibration must be confirmed by measurements!!!!

9. in Figure (4): the legend must be adjusted (Ori !!!!)

10.  in Figure (4): the figure caption must be detailed

11. Figures (5 and 6): the captions must be detailed

12. in Figure (8): the figure caption must be detailed

13. Lines (203-204) must be moved before figure 7

14. Unify the format of figure through the manuscript

15. Lines (219): "In 2.4," must be clarified

16. Lines (274-275): "the dust collection efficiency was calculated by simulating this", the sentence need to be rewritten!!!!

17. Lines (276-280) must be referenced

18. In "Conclusion": what is the benefit of the suggested filters for the applicability

19. All the figures must be enhanced (titles of the axes, symbols, legends, captions)

20. the manuscript need extensive editing and reviewing again

Round 3

Reviewer 2 Report

All figures need enhancement

Author Response

For all the figure and graph, we created a new one and modified the font and theme to match.